# Radiochemical Synthesis and Evaluation of Novel Radioconjugates of Neurokinin 1 Receptor Antagonist Aprepitant Dedicated for NK1R-Positive Tumors

**DOI:** 10.3390/molecules25163756

**Published:** 2020-08-18

**Authors:** Paweł K. Halik, Piotr F. J. Lipiński, Joanna Matalińska, Przemysław Koźmiński, Aleksandra Misicka, Ewa Gniazdowska

**Affiliations:** 1Centre of Radiochemistry and Nuclear Chemistry, Institute of Nuclear Chemistry and Technology, 03-195 Warsaw, Poland; p.kozminski@ichtj.waw.pl (P.K.); e.gniazdowska@ichtj.waw.pl (E.G.); 2Department of Neuropeptides, Mossakowski Medical Research Centre, Polish Academy of Sciences, 02-106 Warsaw, Poland; plipinski@imdik.pan.pl (P.F.J.L.); jmatalinska@imdik.pan.pl (J.M.); misicka@chem.uw.edu.pl (A.M.)

**Keywords:** aprepitant, radiopharmaceuticals, neurokinin 1 receptor antagonist, radionuclide chelators

## Abstract

Aprepitant, a lipophilic and small molecular representative of neurokinin 1 receptor antagonists, is known for its anti-proliferative activity on numerous cancer cell lines that are sensitive to Substance P mitogen action. In the presented research, we developed two novel structural modifications of aprepitant to create aprepitant conjugates with different radionuclide chelators. All of them were radiolabeled with ^68^Ga and ^177^Lu radionuclides and evaluated in terms of their lipophilicity and stability in human serum. Furthermore, fully stable conjugates were examined in molecular modelling with a human neurokinin 1 receptor structure and in a competitive radioligand binding assay using rat brain homogenates in comparison to the aprepitant molecule. This initial research is in the conceptual stage to give potential theranostic-like radiopharmaceutical pairs for the imaging and therapy of neurokinin 1 receptor-overexpressing cancers.

## 1. Introduction

The knowledge of a suitable molecular target and its specificity for a given pathology is a necessary condition in a targeted radionuclide therapy approach. Many malignant tumors possess an infiltrating character with no defined margins or spread out metastases around the whole body. Only the selective binding of a radiopharmaceutical to a molecular target allows for the reliable imaging or safe ablation of cancer lesions with minimal side effects.

Neurokinin 1 receptor (NK1R; tachykinin 1 receptor) is a well-known G protein-coupled receptor for neuropeptide Substance P (SP) and a promising system for an anticancer therapeutic molecular target [1,2]. The activation of the NK1R by its endogenous ligand creates significant proliferative impulses for tumor cells promoting growth and development, including angiogenesis and metastasis. At the same time, the frequent formation of SP-NK1R complexes stimulate the cellular up-regulation of NK1R on tumor cell surfaces [3], thus providing an even greater cell sensitivity for the mitogen action of SP. On the other hand, the blockage of SP action by using antagonists of NK1R on SP-sensitive tumor cells can selectively induce an anti-tumor effect through the mechanism of cell apoptosis [4,5].

Antagonists of NK1R are a very diverse and numerous group of compounds, though clinical applications have only been found for four compounds. They are applied to the prevention of nausea and vomiting induced by chemotherapy or surgical complications [2,6]. One of the best known and widely studied compounds in this group is aprepitant (APT; Figure 1)—a lipophilic and low molecular weight morpholine derivative with a high and selective affinity for NK1R. APT possesses anti-tumor activity, as has been determined in many cancer cell lines [5,7,8,9,10,11,12]. Moreover, the phenomenon of the synergism of the anti-tumor activity of NK1R antagonists with an inhibitory effect on the cancer cell growth of other agents has been confirmed [4]. It has been shown in vitro that the application of microtubule destabilizing agents in combination with antagonists of NK1R possess synergism in apoptotic effect in human glioblastoma, bladder, cervical and breast cancer cells [13]. More remarkable cytotoxic synergism has been proven in a combination of aprepitant and ritonavir (an antiretroviral agent) in the human glioblastoma GAMG cell line [14]. The application of these two drugs with temozolomide, an alkylating chemotherapeutic used clinically to treat glioblastoma, gives an even stronger synergistic effect.

What is most relevant is that APT is a fairly safe drug with a known pharmacological profile, with tolerability similar to placebo- and dose- related action. This could be shown by the fact that aprepitant’s half-maximal inhibitory concentration (IC_50_) value determined for the human embryonic kidney (HEK) 293 cell line (a low expression NK1R control) is higher than the aprepitant IC_100_ values determined for numerous tumor cell lines overexpressing NK1R [15]. For the reasons described above, APT’s structure is an interesting scaffold for creating conjugates for carrying radionuclides to NK1R-positive tumors.

By looking at aprepitant in terms of molecular structure, it can be seen that the compound (Figure 1) consists of a morpholine core decorated by three ‘arms,’ which are:
(i)*p*-fluorophenyl,(ii)3,5-bis-trifluoromethylphenyl suspended at an ether linker, and(iii)a triazolinone moiety suspended at a methylene linker.


In the course of extensive structure-activity studies on NK1R antagonists [16,17,18], it has been established that the first two features (in particular: the distance and mutual positioning of two aromatic rings) are critical for high affinity and, therefore, for NK1R antagonism. On the other hand, the third element, triazolinone ring, can be, at least in some cases, safely modified without a significant loss of affinity [18]. This was exploited in attempts to improve the solubility of aprepitant derivatives, resulting in the derivative L-760,735.

That this site tolerates some modifications is now well-understood in terms of protein-ligand interactions. A recently reported X-ray structure of an NK1R-aprepitant complex [19] revealed that the triazolinone ring is located relatively close to the extracellular end of the receptor binding pocket, where it participates in hydrogen bonding to E193 and W184. However, E193A mutation has virtually no effect on aprepitant’s affinity, thus suggesting that the interactions in this area are of less importance to high affinity binding. Therefore, it seemed the most rational that a convenient site for functionalizing the APT structure is at this very ring. Nevertheless, the performed functionalization of the APT molecule required confirmation that the obtained conjugate still had a sufficiently high affinity for the receptor.

Based on that knowledge, we focused our efforts on the syntheses and in vitro evaluation of newly designed radioconjugates of aprepitant with gallium-68 or lutetium-177 radionuclides. For this purpose, we have proposed two functionalization routes of the APT molecule, followed by conjugation of different macrocyclic chelators DOTA, Bn-DOTA, and Bn-DOTAGA, as well as acrylic chelator DTPA dedicated to ^68^Ga and ^177^Lu. For conjugates showing full stability in human serum, molecular modelling studies for human NK1R and preliminary in vitro examination were performed. These reported findings indicate new perspectives of aprepitant applications in the form of selective theranostic-like concept radiopharmaceuticals for NK1R-positive tumors.

## 2. Results and Discussion

### 2.1. Syntheses of Aprepitant-Based Radioconjugates

#### 2.1.1. Syntheses of Aprepitant Derivatives

The first stage of synthesis concerned the modification of the APT structure in order to introduce a primary amine group. This was realized according to synthetic pathways presented below by using one of selected alkyl linkers (Scheme 1) or acetamide linkers (Scheme 2) so as to receive APT-alkylamine (**2A**–**C**) or APT-acetamide derivatives (**4D**,**E**).

#### 2.1.2. Syntheses of Aprepitant Conjugates

The coupling reactions of APT-ethylamine, **2A**, with different bifunctional chelating agents, were as follows: DOTA-NHS ester, *p*-SCN-Bn-DOTA, *p*-SCN-Bn-DOTAGA, or DTPA dianhydride, as presented in Scheme 3. The use of different chelators allowed for the evaluation of the effect of the chelating moiety on the physicochemical properties of later radioconjugates. Based on the stability results obtained for these radioconjugates (presented in a section below), all other obtained APT derivatives (**2B**, **2C**, **4D**, and **4E**) were only conjugated with selected macrocyclic chelator DOTA. The application of different linkers allowed for the evaluation of their influence on the physicochemical properties of later radioconjugates.

#### 2.1.3. Preparation of Radioconjugates

All APT conjugates with DOTA, Bn-DOTA, and Bn-DOTAGA were radiolabeled with ^68^Ga and ^177^Lu, while APT conjugates with DTPA were only radiolabeled with ^68^Ga. Synthesized radioconjugates were purified using the solid phase extraction (SPE) method before HPLC identification (Figure 2 and Figure 3) and further analyses.

As a result of the performed radiosyntheses, all radioconjugates were successfully obtained, except for [^68^Ga]Ga-DTPA-(Et-APT)_2_ (**[^68^Ga]Ga-9A**), which proved to be immediately unstable. Moreover, in the radiochromatogram of [^177^Lu]Lu-DOTA-Bn-Et-APT (**[^177^Lu]Lu-6A**) one can see a small additional signal (about 19.3 min) that is recognized as an early by-product of an interaction with solvent (EtOH) from the purification process. To verify the identity of all synthesized [^68^Ga]Ga-radioconjugates in a non-carrier added scale, the non-radioactive stable gallium reference compounds (**Ga-5A**–**Ga-9A** and **Ga-5A**–**Ga-5E**) were synthesized and characterized by mass spectrometry. The retention time values of the [^68^Ga]Ga-radioconjugates and stable references presented below (Table 1 and Table 2) overlapped, and the differences between them resulted from the serial connection of UV-Vis and gamma detectors only.

### 2.2. Physiochemical Evaluation of Radioconjugates

#### 2.2.1. Stability Study

The sine qua non condition of a radionuclide’s application in vivo is its radiopharmaceutical stability in biological fluids like serum or cerebrospinal fluid. For this purpose, each isolated and solvent-free radioconjugate was incubated at 37 °C in human serum (HS). At specific time points, small samples of radioconjugate mixture were analyzed by the HPLC method for the assessment of the radioconjugate condition. The collected data presented on the charts below (Figure 4) point out that only the DOTA radioconjugates remained stable in the biological fluid; thus, these radioconjugates were selected for further analyses.

We concluded that for the demand of designed aprepitant radioconjugates, the acyclic chelator DTPA showed a poor radionuclide chelating ability during incubation in human serum. DOTA and its analogues presented a satisfactory radionuclide complex stability, however, for the overall stability of the radioconjugate results from the type of the formed chemical bond with the amine terminated aprepitant derivative and the presence of a negative charge on the chelator-metal complex moiety. The amide bond created by the DOTA-NHS ester and uncharged complex in the conjugates remained stable throughout the whole stability study, while the thiourea bonds and negatively charged complexes created by both *p*-SCN-Bn-DOTA and *p*-SCN-Bn-DOTAGA were found to gradually decompose in time. This phenomenon of instability in HS has been observed previously in various radiopharmaceuticals [20].

#### 2.2.2. Lipophilicity Study

Drug distribution in vivo is highly related to both the lipophilicity and charge of a drug. The optimal radiotracer lipophilicity value for blood-brain barrier crossing lies within the range from 2.0 to 3.5 [21]. Non-peptide NK1R antagonists, like aprepitant, are characterized by a high lipophilicity (logD 4.8) [22], while the DOTA chelator is a highly hydrophilic moiety. In seeking to keep in lipophilicity of radioconjugates in a desired range, the choice of a proper linker (primary aprepitant modification) seems essential for distribution and pharmacokinetic aspects.

In the course of the lipophilicity study, each isolated DOTA radioconjugate (determined as fully stable in HS) was examined for distribution in the system of *n*-octanol and a phosphate-buffered saline (PBS) buffer (pH = 7.4) to estimate the lipophilicity of the radiocomplex. The lipophilicity of each radioconjugate (logD), defined as the logarithm of the distribution coefficient (D) is based on the ratio of the radioactivity of the organic phase to the radioactivity of the aqueous phase. The stability of the studied radioconjugate was verified simultaneously during the experiment through the HPLC analysis of the aqueous phase. LogD values of **[^68^Ga]Ga-5A–[^68^Ga]Ga-5E** and **[^177^Lu]Lu-5A–[^177^Lu]Lu-5E** are listed below in Table 3.

The APT-alkylamine derivative-based radioconjugates showed similar lipophilicity values that were higher than those of the APT-acetamide derivative-based radioconjugates. The complexes with lutetium were more lipophilic by (on average) 0.6 logD units. However, the logD values for all radioconjugates significantly decreased in comparison to aprepitant, indicating possible divergences in the pharmacokinetic fate of the radioconjugates and the parent drug.

### 2.3. Binding Affinity

An important consideration in the search of conjugate vectors for radionuclides is whether the functionalization of a high affinity ligand would not reduce the binding strength for a desired receptor. For the preliminary addressing of this issue in the case of our conjugates, we measured the affinity of compounds **5A**–**E** (uncomplexed precursors) for the rat neurokinin-1 receptor. The human (hNK1R) and the rat (rNK1R) neurokinin 1 receptors differ in their sequences and pharmacology. It has been established that many (but not all) high affinity NK1R antagonists have a significantly lower affinity for the rat receptor than for that of human origin [23,24]. Still, the results presented below give some tentative insight into the affinity changes caused by the functionalization of the aprepitant structure at the triazolinone ring.

The results of the binding affinity determinations are given in Table 4. The parent compound, aprepitant, was found to exhibit IC_50_ = 128.4 nM. This value was roughly consistent with the reported potency of aprepitant in a functional assay. The compound was found to inhibit Substance P-evoked increases in intracellular Ca^2+^ mobilization in the cells expressing rNK1R with a pK_B_ reading 7.3 [25]. Note that in the assays with cells expressing hNK1R, aprepitant was significantly more potent (pK_B_ = 8.7), and the reported binding affinities for the human receptor were of the subnanomolar order (e.g., IC_50_ = 0.09 nM [17]).

The aprepitant-based conjugates exhibited a diversified range of affinities. The strongest ligand in the set was the compound bearing a propylamine linker, **5B**. It was found to have an IC_50_ of 0.69 µM. This value was about five times worse than that of the parent compound. Interestingly, decreasing (**5A**) or increasing (**5C**) the linker length by one methylene unit was associated with much lower affinity of the micromolar order. The shorter **5A** exhibited the lowest binding in the set, with an IC_50_ of 6.2 µM. The analogue with the butylamine linker (**5C**) had an IC_50_ of 1.8 µM. Similar affinities (IC_50_~2.5 µM) were found for the conjugates with the acylhydrazine (**5D**) or *N*-aminoethylacetamide (**5E**) linkers.

### 2.4. Molecular Modelling Study

In order to get insight into possible interactions between the aprepitant-DOTA conjugates reported herein and the NK1R, the complexes thereof were modelled by molecular docking. The applied procedure consisted in building the appropriate linker-DOTA fragments into the aprepitant structure crystallized with the receptor (Protein Data Bank (PDB) accession code: 6HLO [19]), followed by local search docking executed in AutoDock 4.2.6 [26].

According to this procedure, the presence of a linker-DOTA moiety in the aprepitant-based conjugates did not have a major impact on the interactions between the core of the molecule and the receptor. Only a slight repositioning of the morpholine core, 3,5-bis-trifluoromethylphenyl, or *p*-fluorophenyl moieties was observed compared to the 6HLO crystal structure (Figure 5A). Thus, the conjugates were predicted to bind with the 3,5-bis-trifluoromethylphenyl fragment located at the bottom of the ligand-binding pocket and the DOTA moiety closer to the extracellular side of the receptor (Figure 5A).

In the part that was common to all studied derivatives (and the parent aprepitant), the complexes were stabilized by (Figure 5B):
(i)hydrogen bonding between Q165 and the ether oxygen,(ii)hydrophobic contacts of the morpholine ring and F268 and I182,(iii)hydrophobic contacts with side chains of N109, P112, and I113,(iv)hydrophobic contacts of the 3,5-bis-trifluoromethylphenyl with W261 and F264,(v)hydrophobic contacts of the *p*-fluorophenyl ring and H197, V200, T201, I204, and H265.


These interactions were identical to those found for the parent aprepitant in 6HLO structure.

On the other hand, the presence of the linker-DOTA fragment was predicted to weaken the contacts that the triazolinone ring of the parent aprepitant had with the receptor in the crystal structure 6HLO [19]. In the optimized complexes for all the conjugates, this ring was displaced compared to the parent structure (Figure 6A,B), so hydrogen bonding to W184 was not possible. On the other hand, a better positioning of this ring for π–π stacking with H197 was predicted for the conjugates.

Regarding the positioning of the linker-DOTA part, in the case of **5A,** this fragment docked closely (Figure 6C) to the extracellular loop 2 (ECL2) and the extracellular terminus of the transmembrane helix 5 (TM5). One of the DOTA’s carboxylate oxygens interacted with the side-chain of K190. For the analogues **5B**–**D**, the DOTA moiety was predicted to be located between the extracellular tips of TM5 and TM6 (Figure 6D). Its contacts included residues K194, K190, and P271. In the case of the longest derivative, **5E**, the docking placed the DOTA moiety close to TM5 and ECL2 (Figure 6E). Here, it could interact with K190 and M181.

Since in the crystal structures 6HLO, 6HLL, and 6HLP [19], several residues by the extracellular end of the receptor were found to adopt different rotamers upon the binding of different ligands, we wanted to see if the flexibility of these residues could affect the docking results. Therefore, the local docking procedure with the enabled flexibility of E193 and H197 was performed. It yielded similar results with only minor adjustments of the side chain rotamers. Its results (in terms of interactions and binding poses) are not discussed herein since they are almost perfectly accounted for by the description of the docking procedure with the rigid receptor.

Regarding the quantitative evaluation (Table 5), AutoDock scoring function predicted that aprepitant would bind with the free energy of −10.43 kcal/mol. For the conjugates, the estimated energy varied between −9.64 kcal/mol (**5D**) and −13.74 kcal/mol (**5E**). The predicted energies did not correlate with the experimental data. This was perhaps due to the problems with estimating the entropic contribution because the conjugates differed with respect to the number of the rotatable bonds.

Other sources of significant error may have been the way the DOTA moiety was modelled (aimed at mimicking the presence of the cation in a simplified manner) and the fact that the experimentally evaluated conjugates were uncomplexed.

## 3. Materials and Methods

Aprepitant (Santa Cruz Biotechnology Inc., Dallas, TX, USA), the DOTA-NHS ester (1,4,7,10-tetraazacyclododecane-1,4,7,10-tetraacetic acid mono-*N*-hydroxysuccinimide ester), *p*-SCN-Bn-DOTAGA (2,2′,2″-(10-(1-carboxy-4-((4-isothiocyanatobenzyl)amino)-4-oxobutyl)-1,4,7,10-tetraaza-cyclododecane-1,4,7-triyl)triacetic acid) (CheMatech, Dijon, France), *p*-SCN-Bn-DOTA (*S*-2-(4-isothiocyanatobenzyl)-1,4,7,10-tetraazacyclododecane tetraacetic acid) (Macrocyclics, Plano, TX, USA), DTPA dianhydride (diethylenetriaminepentaacetic dianhydride), and other substances and solvents (Sigma Aldrich/Merck, Darmstadt, Germany) were commercially available, defined as reagent grade, and applied without further purification. ^68^GaCl_3_ was eluted from the commercially available ^68^Ge/^68^Ga generator (Eckert & Ziegler, Berlin, Germany). The ^177^LuCl_3_ solution in 0.04 M HCl was purchased at Radioisotope Centre POLATOM, National Centre for Nuclear Research, Otwock-Świerk, Poland. Sep-Pack^®^ Classic Short C18 Cartridges were purchased from WATERS, Milford, MA, USA. Human serum was isolated and purified at the Centre of Radiobiology and Biological Dosimetry, INCT Warsaw, Poland.

The HPLC conditions and gradient were as follows: a semi-preparative Phenomenex Jupiter Proteo column, 4 μm, 90 Å, 250 × 10 mm, with UV/Vis (220 nm) or/and radio γ-detection at gradient elution: 0–20 min 20 to 80% solvent B; 20–30 min 80% solvent B; 2 mL/min; solvent A: 0.1% (*v*/*v*) trifluoroacetic acid (TFA) in water; and solvent B: 0.1% (*v*/*v*) TFA in acetonitrile.

Mass spectra were measured on a Bruker 3000 Esquire mass spectrometer equipped with electrospray ionization (ESI) (Bruker, Billerica, MA, USA).

### 3.1. Syntheses of Aprepitant Derivatives and Aprepitant-Based Conjugates

#### 3.1.1. General Procedure of Syntheses of Aprepitant Derivatives with Alkyl Linker, **2A**–**C**

The slight molar excess of the selected *n*-(terminal-bromoalkyl) phthalimide was added into an equimolar mixture of APT and sodium carbonate in dimethylformamide (DMF). The reaction mixture was vigorously stirred in about 50 °C for 12–18 h. Then, the triple molar excess of hydrazine was added into the reaction mixture for an additional 3 h. The progress of the reaction was monitored by HPLC. The crude reaction mixture was evaporated, dissolved in the HPLC mobile phase, purified by the HPLC method, and lyophilized. The isolated main product was identified as a mono-substituted APT-alkylamine derivative (**2A**–**C**, ~75% reaction yield) by MS analysis confirmation.
MS: Calculated monoisotopic mass for **APT-Et-NH_2_, 2A**, C_25_H_26_F_7_N_5_O_3_: 577.19; found: 578.27 *m*/*z* [M + H^+^]MS: Calculated monoisotopic mass for **APT-Pr-NH_2_, 2B**, C_26_H_28_F_7_N_5_O_3_: 591.21; found: 592.12 *m*/*z* [M + H^+^]MS: Calculated monoisotopic mass for **APT-Bu-NH_2_, 2C**, C_27_H_30_F_7_N_5_O_3_: 605.22; found: 606.38 *m*/*z* [M + H^+^]


#### 3.1.2. General Procedure of Syntheses of Aprepitant Derivatives with Acetamide Linker, **4D** and **4E**

The slight molar excess of ethyl 2-bromoacetate was added into an equimolar mixture of APT and sodium carbonate in DMF. The reaction mixture was vigorously stirred in about 50 °C for 24 h. Then, the triple molar excess of hydrazine or ethylenediamine was added into the reaction mixture for an additional 3 h. The progress of the reaction was monitored by HPLC. The crude reaction mixture was evaporated, dissolved in the HPLC mobile phase, purified by the HPLC method, and lyophilized. The isolated main product was identified as a mono-substituted amino-terminated APT-acetamide derivative (**4D** and **4E**, 65–70% reaction yield) by MS analysis confirmation.
MS: Calculated monoisotopic mass for **APT-Ac-HN-NH_2_, 4D**, C_25_H_25_F_7_N_6_O_4_: 606.19; found: 608.07 *m*/*z* [M + H^+^]MS: Calculated monoisotopic mass for **APT-Ac-Et-NH_2_, 4E**, C_27_H_29_F_7_N_6_O_4_: 634.21; found: 635.31 *m*/*z* [M + H^+^]


#### 3.1.3. General Procedure of Syntheses of Aprepitant Conjugates with DOTA, **5A**–**E**

The obtained APT derivative (**2A**–**C**, **4D**, and **4E**) and the DOTA-NHS ester in similar molar ratios were dissolved in DMF purged from oxygen with technical nitrogen and supplemented with a triple molar excess of triethylamine. The reaction mixture was vigorously stirred in about 50 °C for 24 h. The progress of the reaction was monitored by HPLC. The crude reaction mixture was evaporated, dissolved in the HPLC mobile phase, purified by the HPLC method, and lyophilized. The isolated main product was identified as a DOTA conjugate with an APT derivative (**5A**–**E**, >90% reaction yield) by MS analysis confirmation.
MS: Calculated monoisotopic mass for **APT-Et-DOTA**, **5A**, C_41_H_52_F_7_N_9_O_10_: 963.37; found: 964.27 *m*/*z* [M + H^+^]MS: Calculated monoisotopic mass for **APT-Pr-DOTA**, **5B**, C_42_H_54_F_7_N_9_O_10_: 977.39; found: 978.42 *m*/*z* [M + H^+^]MS: Calculated monoisotopic mass for **APT-Bu-DOTA**, **5C**, C_43_H_56_F_7_N_9_O_10_: 991.40; found: 992.41 *m*/*z* [M + H^+^]MS: Calculated monoisotopic mass for **APT-Ac-HN-NH-DOTA**, **5D**, C_41_H_51_F_7_N_10_O_11_: 992.36; found: 993.17 *m*/*z* [M + H^+^]MS: Calculated monoisotopic mass for **APT-Ac-Et-DOTA**, **5E**, C_43_H_55_F_7_N_10_O_11_: 1020.39; found: 1021.43 *m*/*z* [M + H^+^]


#### 3.1.4. Procedure of Syntheses of Aprepitant-Ethylamine Conjugates with *p*-SCN-Bn-DOTA and *p*-SCN-Bn-DOTAGA, **6A** and **7A**

The APT-ethylamine (**2A**) and bifunctional chelating agent in similar molar ratios were dissolved in DMF and supplemented with a 5-fold molar excess of triethylamine. The reaction mixture was vigorously stirred in about 50 °C for 24 h. The progress of the reaction was monitored by HPLC. The crude reaction mixture was evaporated, dissolved in the HPLC mobile phase, purified by the HPLC method, and lyophilized. The isolated main product was identified as a DOTA-Bn or DOTAGA-Bn conjugate with an APT derivative (**6A** and **7A**, > 90% reaction yield) by MS analysis confirmation.
MS: Calculated monoisotopic mass for **APT-Et-Bn-DOTA**, **6A**, C_49_H_59_F_7_N_10_O_11_S: 1128.40; found: 1129.55 *m*/*z* [M + H^+^]MS: Calculated monoisotopic mass for **APT-Et-Bn-DOTAGA**, **7A**, C_52_H_64_F_7_N_11_O_12_S: 1199.43; found: 1200.66 *m*/*z* [M + H^+^]


#### 3.1.5. Procedure of Syntheses of Aprepitant-Ethylamine Conjugates with DTPA Anhydride, **8A**, **9A**

The APT-ethylamine (**2A**) and DTPA anhydride in a 3:2 molar ratio were dissolved in DMF purged from oxygen with technical nitrogen. The reaction mixture was vigorously stirred in room temperature for 2 h. The progress of the reaction was monitored by HPLC. The crude reaction mixture was evaporated, dissolved in the HPLC mobile phase, purified by the HPLC method, and lyophilized. Two isolated main products were identified as DTPA conjugated with one or two molecules of the APT derivative (**8A** and **9A** with ~45% and ~40% reaction yields, respectively) by MS analysis confirmation.
MS: Calculated for monoisotopic mass **APT-Et-DTPA**, **8A**, C_39_H_47_F_7_N_8_O_12_: 952.32; found: 953.40 *m*/*z* [M + H^+^]MS: Calculated for monoisotopic mass **APT-Et-DTPA-Et-APT**, **9A**, C_64_H_71_F_14_N_13_O_14_: 1511.50; found: 1512.64 *m*/*z* [M + H^+^]


### 3.2. Preparation of Radioconjugates

#### 3.2.1. ^68^Ga Radiolabeling

The ^68^Ga radiolabeling of the DOTA, Bn-DOTA, and Bn-DOTAGA conjugates of APT was performed according to the following procedure: 145 µL of a concentrated solution of [^68^Ga]GaCl_3_ in 0.1 M HCl from the ^68^Ge/^68^Ga generator (4.9 ÷ 7.2 MBq) was added into the solution of 25 nmol of the selected conjugate in 200 µL of a 0.2 M acetate buffer (pH = 4.5) and heated for 5–10 min at 95 °C. After this time, each radioconjugate was purified using Sep-Pack^®^ Classic Short C18 Cartridges according to producer recommendations, thereby obtaining an easily vaporized ethanolic solution of each radioconjugate. The effectiveness of the purification was monitored by HPLC. DTPA radioconjugates were obtained via an analogical procedure in room temperature.

#### 3.2.2. ^177^Lu Radiolabeling

The ^177^Lu radiolabeling of the DOTA, Bn-DOTA, and Bn-DOTAGA conjugates of APT was performed according to the following procedure: 2.7 ÷ 5.3 µL of a [^177^Lu]LuCl_3_ n.c.a. solution in 0.04 M HCl (4.6 ÷ 5.2 MBq) was added into the solution of 2.5 nmol of the selected conjugate in 200 µL of a 0.02 M acetate buffer (pH 4.5) and heated for 10 min at 95 °C. After this time, each radioconjugate was purified using Sep-Pack^®^ C18 Cartridges according to the producer recommendations, thereby obtaining an easily vaporized ethanolic solution of each radioconjugate. The effectiveness of the purification was monitored by HPLC.

#### 3.2.3. Preparation of Non-Radioactive References

The non-radioactive Ga labelling of the DOTA, Bn-DOTA, and Bn-DOTAGA conjugates of APT was performed according to the following procedure: 145 µL of a concentrated solution of 20 mM GaCl_3_ in 0.1 M HCl was added into the solution of 50 nmol of the selected conjugate in 200 µL of a 0.2 M acetate buffer (pH = 4.5) and heated for 5–10 min at 95 °C. After this time, each reaction mixture was purified by the HPLC method, lyophilized, and characterized by mass spectrometry. DTPA conjugates were obtained via an analogical procedure in room temperature.
MS: Calculated for monoisotopic mass **APT-Et-DOTA-Ga**, **5A-Ga**, C_41_H_50_F_7_N_9_O_10_Ga: 1030.80 and 1032.28; found: 1030.40 and 1032.40 *m*/*z* [M^+^]MS: Calculated for monoisotopic mass **APT-Pr-DOTA-Ga**, **5B-Ga**, C_42_H_52_F_7_N_9_O_10_Ga: 1044.30 and 1046.30; found: 1044.38 and 1046.39 *m*/*z* [M^+^]MS: Calculated for monoisotopic mass **APT-Bu-DOTA-Ga**, **5C-Ga**, C_43_H_54_F_7_N_9_O_10_Ga: 1058.31 and 1060.31; found: 1058.37 and 1060.40 *m*/*z* [M^+^]MS: Calculated for monoisotopic mass **APT-Ac-HN-NH-DOTA-Ga**, **5D-Ga**, C_41_H_49_F_7_N_10_O_11_Ga: 1059.27 and 1061.27; found: 1059.31 and 1061.40 *m*/*z* [M^+^]MS: Calculated for monoisotopic mass **APT-Ac-Et-DOTA-Ga**, **5E-Ga**, C_43_H_53_F_7_N_10_O_11_Ga: 1087.30 and 1089.30; found: 1087.37 and 1089.44 *m*/*z* [M^+^]MS: Calculated for monoisotopic mass **APT-Et-Bn-DOTA-Ga**, **6A-Ga**, C_49_H_57_F_7_N_10_O_11_SGa: 1095.31 and 1097.31; found: 1195.54 and 1197.51 *m*/*z* [M^+^]MS: Calculated for monoisotopic mass **APT-Et-Bn-DOTAGA-Ga**, **7A-Ga**, C_52_H_62_F_7_N_11_O_12_Sga: 1266.34 and 1268.34; found: 1266.47 and 1268.47 *m*/*z* [M^+^]MS: Calculated for monoisotopic mass **APT-Et-DTPA-Ga**, **8A-Ga**, C_39_H_44_F_7_N_8_O_12_Ga: 1018.22 and 1020.22; found: 1019.35 and 1021.37 *m*/*z* [M + H^+^]MS: Calculated for monoisotopic mass **APT-Et-DTPA-**(**Ga**)**-Et-APT**, **9A-Ga**, C_64_H_68_F_14_N_13_O_14_Ga: 1577.40 and 1579.40; found: 1578.64 and 1580.66 *m*/*z* [M + H^+^]


### 3.3. Physiochemical Evaluation of Radioconjugates

#### 3.3.1. Stability Study

All obtained radioconjugates (isolated from the reaction mixtures using the SPE method and being solvent-free) were examined in terms of stability in human serum using HPLC analyses. A solution of each isolated selected radioconjugate in 100 µL of a 0.1M PBS buffer pH 7.40 was added to 900 µL of human serum and incubated at 37 °C for 4 h (^68^Ga radioconjugates) or 14 days (^177^Lu radioconjugates). At specific time points, 400 µL of the incubated mixture was added into 500 µL of ethanol, vigorously stirred to precipitate serum proteins, and centrifuged (13,500 rpm for 5 min) to separate the supernatant for HPLC analysis.

#### 3.3.2. Lipophilicity Study

The lipophilicity values of the radioconjugates (logD), expressed as the logarithm of its D in the *n*-octanol/PBS (pH 7.40) system, mimicking the physiological conditions (Product Properties Test Guidelines of the Office of Prevention, Pesticides and Toxic Substances 830.7550, 1996), were determined right after the SPE method purification and ethanol evaporation processes. A solution of isolated selected radioconjugate in 500 µL of a 0.1 M PBS buffer at pH 7.40 and 500 µL of *n*-octanol was vigorously stirred and centrifuged (13,500 rpm for 5 min) to separate the immiscible phases. The radioactivities of the aqueous and organic layers were determined using a well-type NaI(Tl) detector. The distribution coefficient was calculated as the ratio of the radioactivity of the radioconjugate in the organic phase to that in the aqueous phase. Each measurement was performed in triplicate and averaged. Simultaneously, the aqueous phases were analyzed by HPLC to check whether the studied radioconjugate remained intact during the experiment.

### 3.4. Binding Affinity Determination

The binding affinity of aprepitant and compounds **5A**–**E** for rNK1R was determined in a competitive radioligand binding assay using rat brain homogenates, following a previously described method [27]. In brief, the membrane preparations obtained from rat brains were incubated at 25 °C for 60 min in the presence of a selective radioligand [^3^H]-[Sar^9^,Met(O_2_)^11^]-Substance P obtained from PerkinElmer, (Waltham, MA, USA) and the increasing concentrations of the tested compounds (each concentration in duplicate). Non-specific binding was measured in the presence of 10 μM cold Substance P. The assay buffer was composed of 50 mM Tris-HCl (pH 7.4), 5 mM MnCl_2_, bovine serum albumin (BSA) (0.1 mg/mL), bacitracin (100 μg/mL), bestatin (30 μM), phenylmethylsulfonyl fluoride (30 μg/mL), and captopril (10 μM). The reaction total volume was 1 mL. With the incubation having been terminated, a rapid filtration through GF/B Whatman glass fiber strips was done with a M-24 Cell Harvester (Brandel, Gaithersburg, MD, USA). The filters were pre-soaked overnight with 0.5% polyethyleneimine so that the extent of non-specific binding could be minimized. After the filtration, the strips were dried, the filter discs were placed separately in 24-well plates, and a Betaplate Scint scintillation solution (PerkinElmer, Waltham, MA, USA) was added to each well. Radioactivity was measured with a MicroBeta LS scintillation counter, Trilux (PerkinElmer, Waltham, MA, USA). The data came from three independent experiments done in duplicate. The results are presented as IC_50_ with SEM.

### 3.5. Docking

In order to obtain the probable structures of the complexes of the neurokinin 1 receptor with the conjugates **5A**–**E**, the following modelling procedure was performed. The aprepitant structure (with neutral charge) in the complex with the receptor (PDB accession code: 6HLO [19]) was expanded by attaching to the triazolinone ring the appropriate linkers and the DOTA moiety. Such initial complexes were subjected to local search docking in AutoDock 4.2.6 [26].

The DOTA geometry was set based on the NOJYIU entry [28] of The Cambridge Structural Database [29]. This structure is a DOTA complex with Lu^3+^ (diaqua-lutetium(III)-sodium trihydrate). For the purposes of our modelling, DOTA carboxylate arms were protonated and frozen in the conformation found in the crystal structure of lutetium (III) chelate of DOTA (after removing the Lu^3+^ cation, Na^+^ cations and waters). The rationale behind this gambit was the fact that the carboxylates would be primarily engaged in the interactions with a cation; therefore, they might have been expected to retain the conformation they had in the solid state structure. This approach could also give a rough approximation of the DOTA’s steric influence on the binding of the conjugates despite a lack of properly scaled and validated parameters for modelling and scoring the complexes with the cations of interest.

The used receptor structure was a refined one (as provided by the GPCRdb service [30]) in order to have the mutated residues replaced with the native ones and to supply the side chains missing in the original PDB structure. The structure was pre-processed in AutoDock Tools [26]. The box was set around the experimental position of aprepitant in 6HLO and extended towards the extracellular part of the receptor so as to cover the expected length of the expanded conjugate. The grids were calculated with AutoGrid 4 [26]. We considered two variants of docking with respect to the flexibility of the receptor structure. In the first variant, all receptor residues were rigid. In the second variant, E193 and H197 side-chains were set to be flexible.

The docking procedure was the local search with the following parameters: 500 individuals in population, 500 iterations of the Solis-Wets local search, the *sw*_*rho* parameter of the local search space set to 20.0, and 1000 local search runs. The structures resulting from the local search were clustered, and the representative models of the lowest scored (on average) cluster were taken for further analysis. For the qualitative assessment of the binding energy, both the lowest and the mean energy of the clusters were collected. The molecular graphics were prepared in PyMol [31].

For comparative and validation purposes, the very same procedure of local docking (with and without the flexibility of the mentioned two residues) was performed for the parent aprepitant.

## 4. Conclusions

The presented paper describes the evaluation of aprepitant functionalization in order to provide an application of this NK1R antagonist in nuclear medicine.

Out of the corresponding ^68^Ga/^177^Lu radioconjugates of APT-ethylamine **2A** with DOTA, Bn-DOTA, Bn-DOTAGA, and DTPA, only the DOTA amide conjugates showed satisfactory stability in human serum throughout the whole incubation time. The evaluation of the linker effect on radioconjugate lipophilicity indicated APT-alkylamine derivatives as more promising biovectors with features closer to parent aprepitant. The physicochemical properties of obtained APT-alkylamine-DOTA derivatives labelled with ^68^Ga (**[^68^Ga]Ga-5A–[^68^Ga]Ga-5C**) can be compared with those of [^67^Ga]Ga-NOTA-NK1R radioligands based on another NK1R antagonist—L-733,060 [32]. The ^67/68^Ga-radioligands based on these two high affinity NK1R antagonists turned out to be very similar, as evidenced by the following parameters:
(i)they were labelled using macrocyclic chelators (DOTA and NOTA) incorporated in the same ‘arm’ of the antagonist molecule core,(ii)the radioconjugates had similar molecular weights (about 1000),(iii)they had comparable logD values (about 0.15 and 0.6 for the APT-radioligands and the L-733,060-radioligands, respectively),(iv)all were fully stable in human serum examinations.


Regarding the affinity studies of the **5A**–**E** conjugates, on the assumption that the human NK1R affinities for aprepitant derivatives were generally much higher than the rat NK1R affinities and that structure-affinity trends were parallel in both species, all the synthesized compounds might be considered to retain reasonable NK1R affinity compared to their parent. In particular, the analogue **5B** (which only suffered a few-times decrease in affinity compared to APT) seems to be especially interesting for further development. Obtained results suggest that the functionalizing of the aprepitant structure via the triazolinone ring is the right strategy.

It is also worth mentioning that, in general, radiopharmaceuticals based on small non-peptide molecules (e.g., aprepitant and L-733,060) have many advantages over peptide-based radiopharmaceuticals [2]. They usually have lower molecular weights, higher lipophilicity values, and, hence, different pharmacokinetics; they are stable in vivo, but, more importantly, their radiosyntheses can be carried out at higher temperatures and in a wider pH range. Moreover, according to the literature, radiopharmaceuticals based on non-peptide antagonists interact with a receptor through more binding sites and accumulate better and for a longer time period in cancer cells [33,34]. Even though the further evaluation of aprepitant-based radiopharmaceuticals is still needed, the findings reported herein provide insight on the perspectives of their application in the theranostics paradigm.

## 5. Patent

In course of this study, the following national patent application was submitted: No. P430136 “The modified drug substance molecule, method of its production, diagnostic or therapeutic receptor radiopharmaceutical based on this molecule, method of its production and its application”.

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
