# Peer review of "Radiochemical Synthesis and Evaluation of Novel Radioconjugates of Neurokinin 1 Receptor Antagonist Aprepitant Dedicated for NK1R-Positive Tumors"

_molecules, 2020, doi:10.3390/molecules25163756_

Round 1

Reviewer 1 Report

The research proposed for publication is a complex one. In my opinion the paper can be published.

I have only a few remarks:  a list of abbreviations would be welcome,  a correction on line 389: 68Ge/68Ga

Author Response

First of all we would like to thank the Editorial Office and Reviewers for their work done and all the remarks. Below please find answers to the Reviewers' comments in the received order:

Responses to the comments of Reviewer: 1

Comments and Suggestions for Authors:

The research proposed for publication is a complex one. In my opinion the paper can be published. I have only a few remarks:  a list of abbreviations would be welcome,  a correction on line 389: 68Ge/68Ga

Thank you for your comment. A list of abbreviations has been added, however in line 389 upper indexes was prepared originally. We would like to ask for clarification of Reviewers’ request.

Reviewer 2 Report

The manuscript submitted by Halik PK et al. reports the synthesis and evaluation of radiolabeled neurokinin 1 receptor (NK1R)-targeting ligands derived from the antagonist aprepitant. Derivatives with different linkers and chelators were synthesized and radiolabeled with Ga-68 and Lu-177, and evaluated by stability, lipophilicity, and binding assays. Molecular modeling was also conducted to investigate the binding of these chelator-conjugated aprepitant derivatives to NK1R. Although no imaging and/or biodistribution studies were conducted in tumor-bearing mice to directly demonstrate their potential for clinical applications, significant amounts of data were provided, which would be very helpful for use as guidelines for the design of NK1R-targeting radioligands derived from small-molecule inhibitors. Therefore, this manuscript could be accepted for publication after the following suggested changes have been addressed:

  • Line 113: “all other obtained APT derivatives (2A-2C, 4D, 4E) were conjugated only with selected macrocyclic chelator DOTA” should be “all other obtained APT derivatives (2B-2C, 4D, 4E) were conjugated only with selected macrocyclic chelator DOTA”.
  • Figure 2, bottom chromatogram: It seems that there are two Lu-177 labeled DOTA-Bn-Et-APT product peaks (red line). Is this correct? If yes, are they isomers?
  • Line 167: “... while thiourea bonds created by both p-SCN-Bn-DOTA and p-SCN-Bn-DOTAGA were decomposing gradually, most likely due to the action of serum peptidases.”. The authors need to provide examples of what serum peptidases can cleave thiourea bonds.
  • Line 173: “Seeking to keep in radioconjugates as lipophilic as possible ....”. The authors should explain in more detail why they wanted to keep the radioconjugates as lipophilic as possible.
  • Line 194: “... we have measured the affinity of compounds 5A-5E (uncomplexed) for the rat neurokinin-1 receptor.”. The authors should explain why they measured the binding affinity of the precursors, not the nonradioactive Ga-complexed standards.
  • Section 4.1.1: the Authors should provide detailed procedures for the purification of compounds 2A-C.
  • Section 4.1.2: the Authors should provide detailed procedures for the purification of compounds 4D-E.
  • There are two sections of 4.1.2.
  • The authors should provide detailed procedures (and HPLC purification conditions) for the purification of 5A-5E.
  • The authors should provide detailed procedures (and HPLC purification conditions) for the purification of 6A and 7A.
  • The authors should provide detailed procedures (and HPLC purification conditions) for the purification of 8A and 9A.
  • Line 404: “The non-radioactive Ga labelling of all conjugates have been performed in conditions corresponding to described for 68Ga radiolabelling and characterized by mass spectrometry.” This is not an accurate statement as for 68Ga radiolabeling, the 68Ga presented in the reaction mixture is minimal. To prepare the nonradioactive Ga-complexed standards, excess Ga3+ is needed. The authors need to provide detailed procedures (and HPLC purification conditions) for the preparation of nonradioactive Ga-complexed standards.
  • Radiolabeled NK1R-targeting radioligands derived from a small-molecule antagonist L-733060 have been reported by others (Zhang et al. Bioconjugate Chemistry 2018; 29: 1319-1326). The authors should cite this paper and compare their ligands derived from aprepitant with these previously reported radioligands.
  • The same group has previously reported radiolabeled NK1R-targeting ligands derived from peptides. Therefore, the authors should provide rationale for the development of radioligands derived from a small-molecule antagonist, i.e. to compare advantages and disadvantages of radioligands derived from peptides and small molecules.

Author Response

First of all we would like to thank the Editorial Office and Reviewers for their work done and all the remarks. Below please find answers to the Reviewers' comments in the received order:

Responses to the comments of Reviewer: 2

Comments and Suggestions for Authors:

The manuscript submitted by Halik PK et al. reports the synthesis and evaluation of radiolabeled neurokinin 1 receptor (NK1R)-targeting ligands derived from the antagonist aprepitant. Derivatives with different linkers and chelators were synthesized and radiolabeled with Ga-68 and Lu-177, and evaluated by stability, lipophilicity, and binding assays. Molecular modeling was also conducted to investigate the binding of these chelator-conjugated aprepitant derivatives to NK1R. Although no imaging and/or biodistribution studies were conducted in tumor-bearing mice to directly demonstrate their potential for clinical applications, significant amounts of data were provided, which would be very helpful for use as guidelines for the design of NK1R-targeting radioligands derived from small-molecule inhibitors. Therefore, this manuscript could be accepted for publication after the following suggested changes have been addressed:

Thank you for your valuable and carefully prepared comments.

Line 113: “all other obtained APT derivatives (2A-2C, 4D, 4E) were conjugated only with selected macrocyclic chelator DOTA” should be “all other obtained APT derivatives (2B-2C, 4D, 4E) were conjugated only with selected macrocyclic chelator DOTA”.

We have modified the abbreviation according to the Reviewers’ remark.

Figure 2, bottom chromatogram: It seems that there are two Lu-177 labeled DOTA-Bn-Et-APT product peaks (red line). Is this correct? If yes, are they isomers?

That issue was analyzed far by us. In our opinion this can not be an isomer due to following observations:

  • All conjugates with different chelators were made of the same APT-ethylamine derivative 2A batch and only [177Lu]Lu-DOTA-Bn-Et-APT provides two signals;
  • Corresponding radioconjugate, [68Ga]Ga-DOTA-Bn-Et-APT, radiolabeled from the same batch of DOTA-Bn-Et-APT provides single signal;
  • We have investigated recurrence of this radiolabeling and the ratio of the smaller to higher signal varies.

Based on the above, we assume that additional (smaller) peak in [177Lu]Lu-DOTA-Bn-Et-APT radiosynthesis can be rather a very early by-product of interaction with EtOH from SPE purification process. We added a suitable note into the text of article and chart in Figure 4 considering this issue.

Line 167: “... while thiourea bonds created by both p-SCN-Bn-DOTA and p-SCN-Bn-DOTAGA were decomposing gradually, most likely due to the action of serum peptidases.”. The authors need to provide examples of what serum peptidases can cleave thiourea bonds.

Thank you for this remark, we have improved that part of discussion and together with suitable reference.

Line 173: “Seeking to keep in radioconjugates as lipophilic as possible ....”. The authors should explain in more detail why they wanted to keep the radioconjugates as lipophilic as possible.

Thank you for the comment. According to the literature data, the LogD values of radiopharmaceuticals, suitable for crossing the blood-tissue barrier, should be within the range from 1 to 4, and radiopharmaceuticals designed for imaging of the central nervous system, suitable for crossing the blood-brain barrier, should be characterized with LogD value within the range from 2 to 3.5 [Misra, A.; Ganesh, S.; Shahiwala, A.; Shah, S. P. Drug delivery to the central nervous system: a eview. J Pharm Pharm Sci 2003, 6(2), 252-273. PMID: 12935438; Waterhouse, R. N. Determination of lipophilicity and its use as a predictor of blood-brain barrier penetration of molecular imaging agents. Mol Imaging Biol. 2003, 5(6), 376-389. doi:10.1016/j.mibio.2003.09.014; Pike, V. W. PET radiotracers: crossing the blood-brain barrier and surviving metabolism. Trends Pharmacol Sci. 2009, 30(8), 431-440. doi:10.1016/j.tips.2009.05.005]. Since attaching DOTA chelator reduces lipophilicity by 2-3 orders of magnitude, we looked for a linker that would make the conjugate and then the radiotracer as lipophilic as possible. Appropriate information on this issue has been added into the manuscript.

Line 194: “... we have measured the affinity of compounds 5A-5E (uncomplexed) for the rat neurokinin-1 receptor.”. The authors should explain why they measured the binding affinity of the precursors, not the nonradioactive Ga-complexed standards.

Thank you for the remark. We have measured the affinity of compounds 5A-5E (uncomplexed) for the rat neurokinin-1 receptor because the aim of the work at this stage was to provide a tentative insight into the affinity changes caused by functionalization of the aprepitant structure in the triazolinone ring. Currently, the studies of biological properties of 68Ga- and 177Lu-radioconjugates based on 5A-5E precursors are planed using human NK1R-transfected cell lines, however, it will be presented in a future publication.

Section 4.1.1: the Authors should provide detailed procedures for the purification of compounds 2A-C.

Section 4.1.2: the Authors should provide detailed procedures for the purification of compounds 4D-E.

The authors should provide detailed procedures (and HPLC purification conditions) for the purification of 5A-5E.

The authors should provide detailed procedures (and HPLC purification conditions) for the purification of 6A and 7A.

The authors should provide detailed procedures (and HPLC purification conditions) for the purification of 8A and 9A.

We have supplemented the procedures of purification, however they were conducted similarly as it was provided in Material and Method section.

There are two sections of 4.1.2.

Indeed, it is an oversight. Chapter numbering has been corrected in the manuscript.

Line 404: “The non-radioactive Ga labelling of all conjugates have been performed in conditions corresponding to described for 68Ga radiolabelling and characterized by mass spectrometry.” This is not an accurate statement as for 68Ga radiolabeling, the 68Ga presented in the reaction mixture is minimal. To prepare the nonradioactive Ga-complexed standards, excess Ga3+ is needed. The authors need to provide detailed procedures (and HPLC purification conditions) for the preparation of nonradioactive Ga-complexed standards.

Thank you for accurate remark. We have supplemented the whole procedure of cold standard preparation and purification.

Radiolabeled NK1R-targeting radioligands derived from a small-molecule antagonist L-733060 have been reported by others (Zhang et al. Bioconjugate Chemistry 2018; 29: 1319-1326). The authors should cite this paper and compare their ligands derived from aprepitant with these previously reported radioligands.

The mentioned article has been cited in our work and we supplemented the manuscript with a comparison of the aprepitant- and L-733,060-based antagonist radioligands, according to the valuable remark of the Reviewer.

The same group has previously reported radiolabeled NK1R-targeting ligands derived from peptides. Therefore, the authors should provide rationale for the development of radioligands derived from a small-molecule antagonist, i.e. to compare advantages and disadvantages of radioligands derived from peptides and small molecules.

This suggestion has also been followed by us. The manuscript has been supplemented with a short fragment concerning the radiopharmaceuticals based on peptides and small non-peptide antagonists.

Finally, the entire manuscript was carefully checked and minor spelling errors have been corrected.